# DeepWaveRL: Self-Supervised Full Waveform Inversion via Reinforcement Learning

## Abstract

Full Waveform Inversion (FWI) is a fundamental technique to estimate subsurface geophysical properties, such as velocity, from seismic measurements. While supervised deep learning methods have recently shown promising performance by directly mapping seismic data to velocity maps, they require ground-truth velocity maps, which are costly and impractical to obtain at scale. A recent self-supervised approach (UPFWI) removes this dependency by leveraging a differentiable forward operator to reconstruct seismic data from predictions. However, in some practical settings, the forward operator can only be accessed as a black box (e.g., legacy or commercial). Moreover, for complex scenarios, the operator can even be non-differentiable. In this paper, we address this limitation (i.e., the dependency on derivatives of forward operators) by introducing reinforcement learning (RL) into self-supervised FWI. Our method, named DeepWaveRL, reformulates FWI as a policy learning problem, where the model generates velocity maps as actions, and the forward operator is used only to compute rewards. This design avoids backpropagation through the forward operator, thus eliminating the need to compute its derivatives. Furthermore, we identify key strategies to stabilize reinforcement learning in this challenging setting. In the absence of ground-truth labels and differentiable forward operators, our method achieves competitive performance compared to supervised counterparts. We believe our approach provides a more flexible solution for the FWI research community.

## 1 Introduction

Subsurface imaging is essential for characterizing geological structures and geophysical properties (e.g., velocity and impedance), with applications in energy exploration, carbon capture and sequestration, and earthquake early warning systems. A central technique in this domain is Full Waveform Inversion (FWI), which estimates subsurface velocity maps from seismic measurements. Typically, seismic data are acquired through seismic surveys, where an array of receivers records reflected and refracted seismic waves. These waves are generated by controlled sources. Mathematically, for an isotropic medium with constant density, the velocity map and seismic measurements are connected by the acoustic wave equation:

$$\nabla^2 p(x, z, t) - \frac{1}{v(x, z)^2} \frac{\partial^2 p(x, z, t)}{\partial t^2} = s(x, z, t) ,\qquad(1)$$

where $x$ denotes the horizontal offset, $z$ the depth, $p(x, z, t)$ the pressure wavefield at spatial location $(x, z)$ and time $t$, $v(x, z)$ the wave propagation velocity at $(x, z)$, $s(x, z, t)$ the source term, and $\nabla^2$ the Laplacian operator. In practice, seismic data are often collected at the surface (i.e., $p(x, z = 0, t)$). While FWI has the potential to produce high-resolution velocity maps, the inverse problem itself is inherently non-linear and ill-posed. In addition, conventional physics-driven approaches face additional challenges, as they require intensive computation due to repeated forward simulations per sample and exhibit strong sensitivity to noise and initial conditions. These challenges have motivated growing interest in data-driven deep learning methods.

A majority of data-driven methods (Wu & Lin, 2019; Zhang et al., 2019; Jin et al., 2024) adopt a supervised learning paradigm and formulate FWI as an image-to-image translation task. As shown in Figure 1, deep neural networks are trained to directly learn the mapping from seismic data to

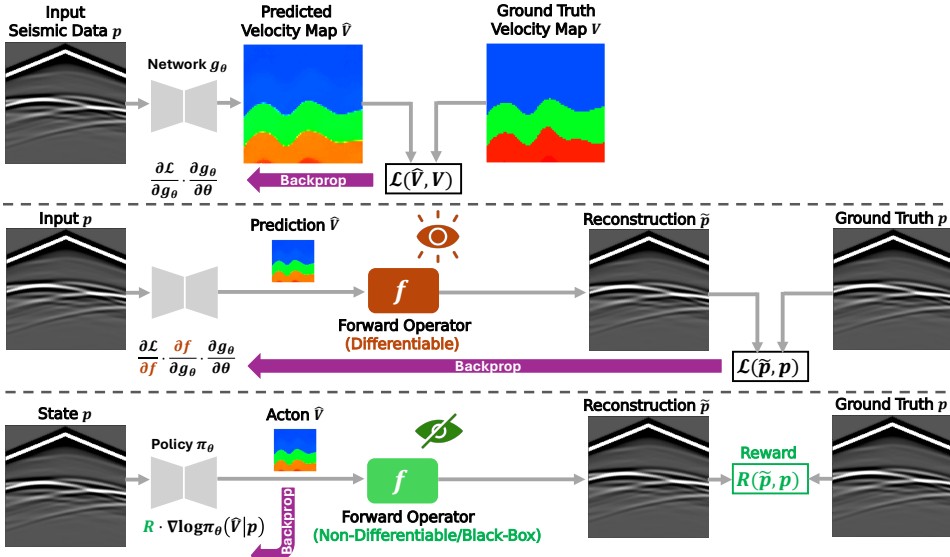

Figure 1: Comparison between different data-driven FWI methods. **Top**: Supervised learning methods compute loss between predicted and ground-truth velocity maps; **Middle**: Self-supervised learning method with a differentiable forward operator $f$ computes the loss between input and reconstructed seismic data and backpropagates gradients through $f$; **Bottom**: Our proposed DeepWaveRL uses the forward operator only to compute a reward signal based on misfit between seismic data, without backpropagating gradients through $f$, enabling greater flexibility.

velocity maps, enabling fast inference and achieving high accuracy under ideal conditions. However, these methods rely on a large amount of paired seismic data and velocity maps for training. In real-world scenarios, such ground-truth velocity maps are rarely available because constructing them is extremely time-consuming and requires substantial expertise from geophysicists.

A recent work (UPFWI, Jin et al., 2022) explicitly leverages the underlying physics knowledge and achieves self-supervised learning without ground-truth velocity maps. As illustrated in Figure 1, a *differentiable* forward operator $f$ is coupled with a neural network to simulate seismic data from predicted velocity maps. By minimizing reconstruction loss on seismic data with *gradients back-propagated through $f$*, the network can be trained in an end-to-end manner without labeled supervision. However, this design imposes several critical limitations. First, the need for differentiability restricts the choice of forward solvers: many high-performance seismic simulators are implemented in low-level languages such as Fortran or C++ and are only available as non-differentiable "black boxes." Second, real physical systems often exhibit non-smooth behaviors—for example, fractures that open only beyond a pressure threshold—where the wavefield response can change abruptly, violating differentiability and further limiting the applicability of such approaches.

In this paper, we present DeepWaveRL, a novel self-supervised approach for FWI that removes the dependency on differentiable forward operators by leveraging reinforcement learning (RL). As depicted in Figure 1, we formulate FWI as a single-step decision problem, where the input seismic data serve as the state, and a policy network outputs the corresponding velocity map as the action. A forward operator $f$ is still used, but *only to compute rewards* based on the simulated seismic data from the predicted velocity map. The policy is then optimized via a policy gradient algorithm, which computes the gradients of the network's action probabilities, weighted by the reward signal. Therefore, our method *eliminates the need to backpropagate gradients through $f$*.

Policy optimization in this setting poses unique challenges: continuous velocity values lead to an enormous action space, hindering effective exploration; large amplitude disparities between waves bias learning toward dominant ones; reward signals can only be evaluated for the entire velocity map without pixel-level feedback.

To address these issues, we further identify three key strategies. First, we use a discrete action space by partitioning the velocity range into finite bins, which significantly reduces the burden of exploration while maintaining accuracy. Second, we adopt a sign-preserving logarithmic transformation

for seismic data that compresses dominant directive wave energy and amplifies weaker signals (e.g., reflections and deep arrivals), thereby yielding more precise predictions in deeper regions. Third, we exploit the ability of a well-trained policy to adapt across datasets, allowing transfer of knowledge in scenarios where training from scratch would be difficult or unstable.

We evaluate our method on several datasets from OpenFWI (Deng et al., 2022), a large-scale, multi-structural dataset collection. Experimental results show that our DeepWaveRL attains comparable performance to the supervised baseline InversionNet (Wu & Lin, 2019; Jin et al., 2024) on CurveVel-A, with a Mean Absolute Error (MAE) of 0.0527 (vs. 0.0409), a Root Mean Squared Error (RMSE) of 0.1012 (vs. 0.0944), and a Structured Similarity (SSIM) of 0.8601 (vs. 0.8796). DeepWaveRL with transfer learning also yields competitive performance on FlatFault-A and CurveFault-A.

Our contribution is summarized as follows:

- We propose DeepWaveRL, a reinforcement learning framework for self-supervised full waveform inversion (FWI), which removes the need for differentiable forward operators.
- We propose three key techniques for stable and efficient training of DeepWaveRL, including discretized velocity actions, sign-preserving logarithmic transformation on seismic data, and transfer learning of well-trained policies.
- We demonstrate that our proposed DeepWaveRL achieves competitive performance without the involvement of ground-truth labels and differentiable forward operators.

## 2 METHOD

In this section, we first briefly summarize the state-of-the-art group-based reinforcement learning algorithms and then present our DeepWaveRL and its components. After that, we provide a summary of the comparison of DeepWaveRL with previous FWI methods from a gradient perspective.

### 2.1 PRELIMINARY

Shao et al. (2024) introduces Group Relative Policy Optimization (GRPO) that enhances Proximal Policy Optimization (PPO, Schulman et al., 2017) by omitting the value function and estimating the advantage in a group-relative manner. This is followed by several variants such as Decoupled Clip and Dynamic Sampling Policy Optimization (DAPO, Yu et al., 2025) and Group Sequence Policy Optimization (GSPO, Zheng et al., 2025), yielding even superior training efficiency and performance. The core idea of GRPO is summarized as follows.

For a specific question $q \sim P(Q)$, a group of $G$ responses $\{o_i\}_{i=1}^G$ are sampled from an old policy network $\pi_{\theta_{\text{old}}}$. Each response $o_i$ is then fed into a reward function to obtain the individual reward $R_i$. By normalizing the rewards within each group, an advantage $\hat{A}_i$ is assigned to each response. The policy network is optimized by maximizing the following clipped objective, similar to PPO:

$$\mathcal{J}_{\text{GRPO}}(\theta) = \mathbb{E}_{q \sim P(Q), \{o_i\}_{i=1}^G \sim \pi_{\theta_{\text{old}}}(\cdot|q)}$$
$$\frac{1}{G} \sum_{i=1}^G \frac{1}{|o_i|} \sum_{t=1}^{|o_i|} \left\{ \min \left[ r_{i,t}(\theta)\hat{A}_{i,t}, \text{clip}\left( r_{i,t}(\theta), 1 - \varepsilon, 1 + \varepsilon \right)\hat{A}_{i,t} \right] - \beta D_{\text{KL}}(\pi_\theta || \pi_{\text{ref}}) \right\},$$

$$(2)$$

where $\epsilon$ is the hyperparameter to determine the clipping boundaries, and $\beta$ is to control the importance of the KL divergence $D_{\text{KL}}$ between the online policy $\pi_\theta$ and the frozen reference policy $\pi_{\text{ref}}$. In addition, $\hat{A}_{i,t}$ and $r_{i,t}(\theta)$ are the group-based advantage estimation and importance ratio of $o_{i,t}$, which is the $t-$th token in response $o_i$. They are defined as:

$$\hat{A}_{i,t} = \hat{A}_i = \frac{R_i - \text{mean}(\{R_i\}_{i=1}^G)}{\text{std}(\{R_i\}_{i=1}^G)}, \quad r_{i,t}(\theta) = \frac{\pi_\theta(o_{i,t} \mid q, o_{i,<t})}{\pi_{\theta_{\text{old}}}(o_{i,t} \mid q, o_{i,<t})}. \quad (3)$$

### 2.2 SELF-SUPERVISED FWI VIA REINFORCEMENT LEARNING

As illustrated in Figure 2, DeepWaveRL formulates FWI as a single-step policy learning problem. Each input seismic data $p \in \mathbb{R}^{N_s \times N_t \times N_r}$ is treated as a state and passed into the policy network $\pi_\theta$.

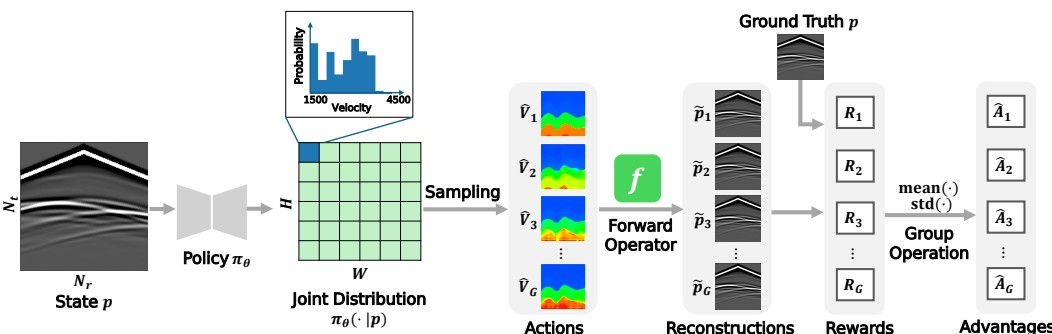

Figure 2: Schematic illustration of the policy optimization pipeline in DeepWaveRL. For brevity, we only show the seismic data from one source.

Here, $N_s$ denotes the number of sources used during data acquisition, $N_t$ the number of recorded timesteps, and $N_r$ the total number of receivers. The network then outputs the probability distribution of velocity $\pi_\theta(\cdot|p)_{h,w}$ at each spatial location $(h, w)$. Following GRPO, we sample a group of $G$ velocity maps $\{\hat{V}_i\}_{i=1}^{G}$ from the predicted joint probability distribution as 2D actions, where each $\hat{V}_i = \{\hat{v}_{i,h,w}\}_{h=1,w=1}^{H,W}$, and $H$ and $W$ are the vertical and horizontal dimensions of the velocity map. To reduce exploration burden, we use a discrete action space instead of the continuous one.

To assign a reward $R_i$ to each sampled action $\hat{V}_i$, we employ a forward operator $f$ to simulate seismic data $\tilde{p}_i = f(\hat{V}_i) \in \mathbb{R}^{N_s \times N_t \times N_r}$. The reward is then computed based on the misfit between the input seismic data $p$ and the reconstruction $\tilde{p}_i$. We further compute the relative advantage $\hat{A}_i$ within each group following Equation 3, but use a map-level importance ratio.

According to a recent work (Zheng et al., 2025), the mismatch between the unit of reward and the unit of optimization objective can introduce high-variance noise and further lead to model collapse. In our settings, a reward is assigned to the whole 2D velocity map. Therefore, instead of computing the pixel-level importance ratios in Equation 3, we define the map-level importance ratio as:

$$m_i = \left[\frac{\pi_\theta(\hat{V}_i|p)}{\pi_{\theta_{old}}(\hat{V}_i|p)}\right]^{\frac{1}{H \cdot W}} = \exp\left[\frac{1}{H \cdot W}\sum_{h=1}^{H}\sum_{w=1}^{W}\log\left(\frac{\pi_\theta(\hat{v}_{i,h,w}|p)}{\pi_{\theta_{old}}(\hat{v}_{i,h,w}|p)}\right)\right]. \quad (4)$$

Consequently, the overall optimization objective can now be written as:

$$\mathcal{J}(\theta) = \mathbb{E}_{p\sim\mathcal{P},\{\hat{V}\}_{i=1}^{G}\sim\pi_{\theta_{old}}(\cdot|p)}$$
$$\frac{1}{G}\sum_{i=1}^{G}\left\{\min\left[m_i(\theta)\hat{A}_i, \text{clip}\left(m_i(\theta), 1-\varepsilon_{low}, 1+\varepsilon_{high}\right)\hat{A}_i\right]\right\}, \quad (5)$$

where we follow Yu et al. (2025) and decouple the lower and higher clipping range as $\varepsilon_{low}$ and $\varepsilon_{high}$, and $\mathcal{P}$ denotes the distribution of seismic data. We also remove the KL penalty term as our initial model is not as good as common pretrained language models, and we allow the model distribution to diverge from the initial model.

The policy network is thus trained to shift its output distribution toward higher-reward actions. Importantly, the entire training process is self-supervised, without the involvement of ground-truth velocity maps. Furthermore, no gradients are backpropagated through the forward operator $f$, which allows $f$ to be arbitrary, including non-differentiable or black-box simulators.

Additionally, we note that the policy network can also be optimized at test time, since only seismic data are required. This test-time optimization further boosts performance, which will be discussed in Section 3.3.

### 2.3 KEY STRATEGIES FOR STABLE AND EFFICIENT LEARNING

**Discrete Action Space.** During training, we treat the predicted velocity map $\hat{V} = \{\hat{v}_{h,w}\}_{h=1,w=1}^{H,W}$ as a 2D action sampled from the joint distribution. We initially experimented with a continuous action

space, where the network outputs two maps of size $H \times W$, representing the mean $\mu_{h,w}$ and standard deviation $\sigma_{h,w}$ of a Gaussian distribution at each spatial location. Hence, the velocity at location $(h, w)$ was sampled as $\hat{v}_{h,w} \sim \mathcal{N}(\mu_{h,w}, \sigma_{h,w})$. However, this formulation led to unstable training and noisy predictions. To address these issues, we discretize the velocity values uniformly into $B$ bins, treating the sampled action as a predicted category. Given an action $a_{h,w} \in [0, 1, ..., B-1)$, we compute the corresponding velocity as

$$v(a_{h,w}) = \frac{v_{\max} - v_{\min}}{B} \cdot (a_{h,w} + 0.5) + v_{\min}, \qquad (6)$$

where $v_{\max}$ and $v_{\min}$ are the largest and smallest possible velocities in a dataset. This discrete action space substantially improves both training stability and prediction quality.

**Sign-Preserving Logarithm Transformation in Reward.** Similar to the loss function of UP-FWI (Jin et al., 2022), we define the reward as the negative pixel-wise $\ell_1$ and $\ell_2$ distance between the input and reconstructed seismic data. To further enhance learning, we apply a sign-preserving logarithm transformation during the computation of rewards as:

$$p' = \text{sign}(p) \cdot \log(k \cdot |p| + c), \qquad (7)$$

where $k$ and $c$ are hyperparameters to control the strength of the transformation. This non-linear transformation can compress dominant directive wave energy and amplify weaker signals (e.g., reflections and deep arrivals), thereby guiding the network to recover more accurate velocities in deeper regions. The reward function can then be described as:

$$R = -\ell_1(p', \tilde{p}') - \ell_2(p', \tilde{p}'). \qquad (8)$$

**Transfer Learning using Well-Trained Policies.** Directly training our DeepWaveRL from scratch sometimes leads to unstable learning and convergence issues. We provide examples of predicted velocity maps generated by these models in Appendix D.5. The predictions exhibit unrealistically low velocities in deep regions after certain training steps. To address this issue, we propose to initialize the policy network with weights from a well-trained model on a different dataset. This transfer learning strategy allows prior knowledge of velocity distributions to be reused across datasets, leading to more stable training and improved convergence.

## 2.4 Comparison with Previous FWI Methods from a Gradient Perspective

To demonstrate the relationship among supervised, self-supervised with a differentiable forward operator, and RL-based self-supervised approaches, we provide analysis from the perspective of gradient construction and propagation.

**Supervised:** Gradients are directly computed from discrepancies with ground-truth velocity maps, $\nabla_\theta \mathcal{L}_{\text{sup}} \propto \frac{\partial \| V - \hat{V} \|}{\partial \hat{V}}$. While this yields stable optimization and strong supervision, it is impractical in real-world scenarios due to the scarcity of paired data.

**Differentiable self-supervised:** Gradients originate from seismic reconstruction error and back-propagate through a differentiable forward operator, $\nabla_\theta \mathcal{L}_{\text{diff}} \propto \frac{\partial f(\hat{V})}{\partial \hat{V}}$, enabling physics-informed learning but restricted by differentiability and high computational cost.

**RL-based self-supervised:** In contrast to self-supervised methods with a differentiable forward operator, where the loss must be differentiable with respect to $\hat{V}$ and thus optimization is tightly coupled to $f$, our RL-based method replaces this requirement by converting non-differentiable errors into reward signals that act as multipliers of the policy gradients, $\nabla_\theta \mathcal{J}(\theta) \propto \mathbb{E}\left[\hat{A} \cdot \nabla_\theta \log m_i(\theta)\right]$.

Full derivations and detailed comparisons are provided in Appendix B.

## 3 Experiments

In this section, we evaluate the performance of our proposed DeepWaveRL on OpenFWI (Deng et al., 2022), comparing it with both supervised and self-supervised baselines. We also investigate the impact of the discrete action space and examine the effect of the logarithm transformation on seismic data through ablation studies.

Table 1: Quantitative results evaluated on CurveVel-A. For models with discrete predictions, we report the mean estimate.

| Method | MAE↓ | RMSE↓ | SSIM↑ |
|---|---|---|---|
| InversionNet | **0.0409** | **0.0944** | **0.8796** |
| UPFWI | 0.0805 | 0.1411 | 0.8443 |
| DeepWaveRL | 0.0717 | 0.1300 | 0.8303 |
| DeepWaveRL + TTO | 0.0527 | 0.1012 | 0.8601 |

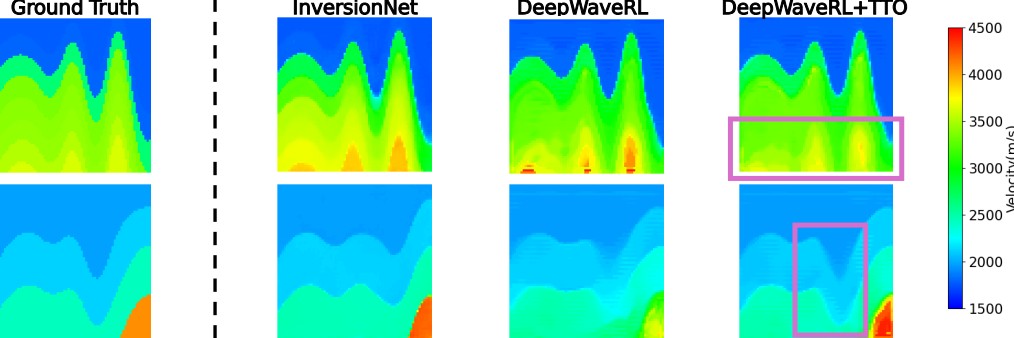

Figure 3: Illustration of ground truth and inversion results of different methods on CurveVel-A.

## 3.1 DATASETS

We verify our method on CurveVel-A, FlatFault-A and CurveFault-A of OpenFWI (Deng et al., 2022), an open-source collection of large-scale, multi-structural benchmark datasets for data-driven seismic FWI. CurveVel-A contains velocity maps composed of curved layers with clear interfaces, while FlatFault-A and CurveFault-A focus more on geological fault identification and have velocity maps with flat and curved layers, respectively. CurveVel-A contains 30K velocity maps and their corresponding seismic data. Following the official data split, we use 24K for training and 6K for testing. FlatFault-A and CurveFault-A contain 54K samples each, and we use 48K/6K splitting.

Each velocity map in all three datasets has a size of $70 \times 70$, with a grid size of 10 meters in both horizontal and depth directions. The velocity value ranges from 1,500 meter/second to 4,500 meter/second. For seismic data, five sources are placed evenly with a 170-meter spacing and a central source frequency of 15 Hz. The seismic data are recorded by 70 receivers at 10-meter intervals, each collecting 1,000 timesteps over 1 second. This results in seismic data of shape $5 \times 1000 \times 70$. For additional details, we refer readers to the original OpenFWI paper (Deng et al., 2022).

## 3.2 EXPERIMENT SETTINGS

**Evaluation Metrics:** We evaluate predicted velocity maps using MAE, RMSE, and Structural Similarity (SSIM), consistent with prior work (Wu & Lin, 2019; Feng et al., 2024; Deng et al., 2022). MAE and RMSE quantify pixel-wise errors, while SSIM captures perceptual similarity, reflecting the structured information of velocity maps where distortions can be easily perceived by a human. Note that all measurements are computed on normalized velocity maps, with MAE and RMSE in the range $[-1, 1]$, and SSIM in $[0, 1]$.

**Comparison:** We compare our method to InversionNet (Wu & Lin, 2019) which achieves the state-of-the-art performance when trained solely on each dataset, as demonstrated in a recent work (Jin et al., 2024). Additionally, we list the benchmarking results of UPFWI (Jin et al., 2022) from the original OpenFWI paper.

Technical details regarding training are provided in Appendix C.

Table 2: Quantitative results evaluated on FlatFault-A and CurveFault-A.

| Dataset | Method | MAE↓ | RMSE↓ | SSIM↑ |
|---|---|---|---|---|
| FlatFault-A | InversionNet | **0.0098** | **0.0276** | **0.9880** |
| | UPFWI | 0.0876 | 0.2060 | 0.9340 |
| | DeepWaveRL | 0.0301 | 0.0557 | 0.9062 |
| | DeepWaveRL + TTO | 0.0268 | 0.0476 | 0.9146 |
| CurveFault-A | InversionNet | **0.0164** | **0.0480** | **0.9721** |
| | UPFWI | 0.0500 | 0.0966 | 0.9495 |
| | DeepWaveRL | 0.0362 | 0.0703 | 0.9111 |
| | DeepWaveRL + TTO | 0.0278 | 0.0502 | 0.9323 |

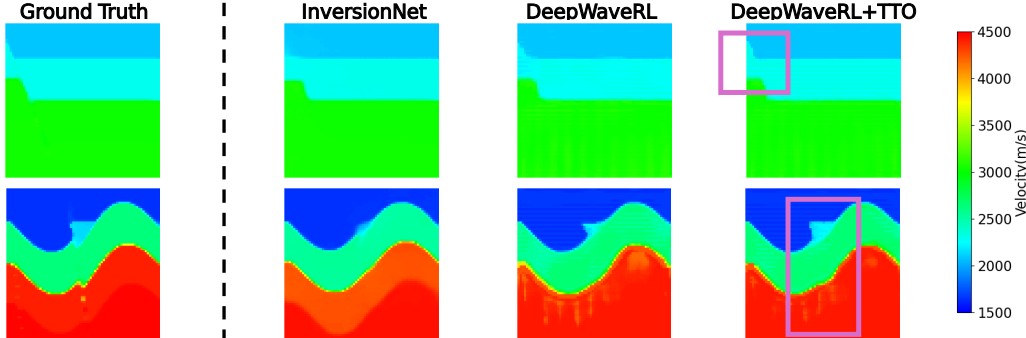

Figure 4: Illustration of ground truth and inversion results of different methods on FlatFault-A (top) and CurveFault-A (bottom).

## 3.3 MAIN RESULTS

**Results on CurveVel-A:** Table 1 shows the quantitative results of different methods on CurveVel-A. For the models that predict velocity as discrete values, we report the mean prediction (expected value) where we compute the expectation over the bin values using their predicted probabilities.

Among all the models, InversionNet yields the best performance, which is expected as it leverages supervised learning and directly predicts continuous velocity values. In comparison, our DeepWaveRL with test-time optimization (DeepWaveRL+TTO) attains the second-best performance with a slight gap. Notably, test-time optimization substantially boosts the performance of DeepWaveRL.

Figure 3 further illustrates examples of ground-truth velocity maps and inversion results from different methods. Here, the results of DeepWaveRL models are all mean predictions. Consistent with our quantitative analysis, InversionNet produces sharp layer boundaries and smooth and uniform velocities within layers. However, in certain regions, DeepWaveRL+TTO yields more details. For instance, as highlighted in the first row, the predictions of DeepWaveRL+TTO have more accurate velocities in deep regions, whereas InversionNet introduces additional layers and predicts inaccurate velocities. Another observation is that only DeepWaveRL+TTO precisely reconstructs the subsurface structure in the highlighted areas in the second row. Moreover, when comparing the results of DeepWaveRL and DeepWaveRL+TTO, we find that test-time optimization helps eliminate artifacts, recover curved structures, and improve the accuracy of intra-layer velocities. More visualization results are shown in Appendix D.1.

**Transfer Learning Results on FlatFault-A and CurveFault-A:** Table 2 lists the quantitative results on FlatFault-A and CurveFault-A, and mean predictions are reported. For transfer learning, we adopt the last checkpoint of the DeepWaveRL model trained with test-time optimization on CurveVel-A. For both datasets, our DeepWaveRL+TTO consistently outperforms UPFWI in terms of MAE and RMSE, with small gaps in SSIM. This indicates that DeepWaveRL-TTO yields generally accurate predictions, but there may be small shifts or artifacts that are visually noticeable.

Table 3: Quantitative results evaluated on CurveVel-A, with different choices of action space.

| Setting | MAE↓ | RMSE↓ | SSIM↑ |
| --- | --- | --- | --- |
| Continuous Action Space | 0.1443 | 0.2073 | 0.4359 |
| Discrete Action Space | 0.0527 | 0.1012 | 0.8601 |

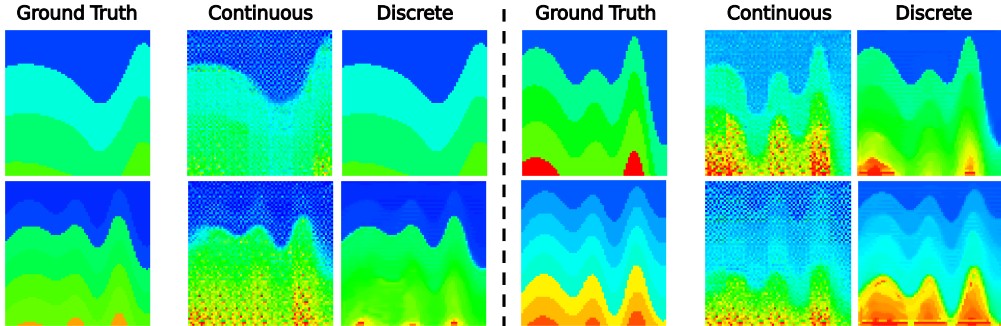

Figure 5: Illustration of ground truth and inversion results of DeepWaveRL with different choices of action space

The visualization results in Figure 4 further support our hypothesis. Despite some stripe artifacts, our DeepWaveRL and DeepWaveRL+TTO generate more precise details in some shallow regions. As highlighted in the first row (FlatFault-A), DeepWaveRL models reconstruct the fault on the top-left corner, while it is barely visible in the predictions of InversionNet. Similarly, in the second row (CurveFault-A), our DeepWaveRL models precisely capture the triangle-shaped region. More visualization results are shown in Appendix D.2.

## 3.4 ABLATIONS

**Continuous vs. Discrete Action Space:** We further analyse how the choice of action space affects model performance. The quantitative results are summarized in Table 3. For simplicity, we denote DeepWaveRL with a discrete action space as DeepWaveRL-D, and with a continuous action space as DeepWaveRL-C. In terms of all three metrics, DeepWaveRL-D outperforms DeepWaveRL-C to a large extent. The gap is particularly large in SSIM, suggesting that DeepWaveRL-C fails to produce results consistent with human perceptual quality.

Figure 5 provides qualitative comparisons between DeepWave-C and DeepWaveRL-D, which further support our quantitative analysis. While DeepWaveRL-C can recover some structures in shallow regions, there are plenty of artifacts all over the predictions. In particular, the high-velocity areas are severely corrupted, making boundaries unrecognizable. Furthermore, these artifacts persist even when training is extended, indicating that the continuous action space poses significant optimization challenges. By contrast, discretization reduces the complexity of the action space and greatly stabilizes training, leading to more accurate and reliable results.

**With or Without Logarithm Transformation:** To evaluate the effect of the sign-preserving logarithm transformation, we train our DeepWaveRL without transformation on CurveVel-A. The resulting MAE, RMSE, and SSIM are 0.0785, 0.1383, and 0.8195, respectively. Compared to the results of DeepWaveRL with transformation in Table 1, the performance degrades in terms of all three metrics. This is consistent with the visualization results in Appendix D.3, where some details are missing in deep regions.

## 4 DISCUSSION

During qualitative analysis, we find that mode collapse occurs in some of the predictions of our DeepWaveRL, as illustrated in Appendix D.4. The ground-truth velocity maps of these predictions have close velocities in their shallow regions, but this pattern does not guarantee the occurrence of

mode collapse. Thus, we may take this into consideration in our future work. Another limitation is that our DeepWaveRL still struggles with the recovery of the structures in deeper regions due to the inherent attenuation of signals in these regions. Our sign-preserving logarithm transformation is one of the solutions, but more advanced algorithms are still needed. Furthermore, our DeepWaveRL framework enables another potential research direction, which is to incorporate non-differentiable regularization terms such as total variation into the reward design.

## 5 RELATED WORK

**Deep Learning for FWI:** Deep learning approaches to FWI span data-driven, physics-informed, and hybrid paradigms (Lin et al., 2023; Adler et al., 2021; Yu & Ma, 2021). Fully supervised methods (Araya-Polo et al., 2018; Wu & Lin, 2019; Zhang et al., 2019; Li et al., 2020) learn direct mappings from seismic data to velocity models using paired data, which are costly to acquire and often lead to poor generalization under domain shifts. To reduce reliance on labels and improve robustness, self-supervised strategies have emerged. Feng et al. (Feng et al., 2022; 2024) decouple the seismic encoder and velocity decoder by leveraging latent space correlations, enabling separate training. SiameseFWI (Saad et al., 2024) explores self-supervision with a Siamese network that better aligns simulated and observed data. Semi-supervised learning has also been explored. Sun et al. (2023) proposes a CycleGAN-based framework to reconstruct missing low-frequency components in field data. Other methods generate pseudo-labels from unlabeled or auxiliary data (Rojas-Gómez et al., 2022; Cai et al., 2022), bridging the gap between labeled and unlabeled domains. Unsupervised methods such as UPFWI (Jin et al., 2022) and Jia et al. (Jia et al., 2025) go further by eliminating labels entirely. These approaches minimize waveform mismatches under physical constraints, using differentiable forward modeling to optimize predicted velocity maps. However, their reliance on computationally intensive and differentiable solvers limits scalability to high-resolution, elastic, or 3D FWI, and precludes use with black-box simulators. Diffusion models offer an alternative by learning generative priors that guide the inverse process via plug-and-play (PnP) denoising (Song et al., 2022; Chung et al., 2023; Zhang et al., 2025). Wang et al. (2023) successfully applies this strategy to FWI. While these models avoid paired supervision, they still require large velocity datasets for training and incur high inference costs due to repeated forward simulations during sampling.

**Reinforcement learning and group-based policy optimization:** Policy-gradient RL methods (e.g., PPO (Schulman et al., 2017)) provide a principled way to optimize stochastic policies via likelihood-ratio estimators, and have been widely used in sequential generation and control. Recent advances in group-/sequence-level policy optimization demonstrate that performing importance-weighting and clipping at the unit-of-reward level (group or sequence) can reduce variance and stabilize training for structured outputs. In particular, DeepSeekMath (Shao et al., 2024) replaces value-function estimation with group-relative normalization, yielding more efficient and stable updates. They show that this framework can be scaled to mathematical reasoning tasks with strong generalization. DAPO (Yu et al., 2025) extends the paradigm by decoupling clipping ranges and introducing dynamic sampling, further improving stability under diverse reward distributions. GSPO (Zheng et al., 2025) generalizes these ideas to full sequence-level optimization with the importance ratio based on sequence likelihood, aligning long-horizon objectives with token-level policies. Inspired by this progression, DeepWaveRL adapts the same philosophy to the geophysics domain by treating an entire velocity map as a structured action, making map-level optimization a natural extension of group-based RL methods for physics-driven inverse problems.

## 6 CONCLUSION

In this study, we introduce DeepWaveRL, a reinforcement learning framework for self-supervised full waveform inversion that eliminates the need for differentiable forward operators. By incorporating discretized velocity actions, a sign-preserving logarithmic transformation of seismic data, and transfer learning from well-trained policies, DeepWaveRL achieves stable and efficient training. We demonstrate through experiments that our method attains competitive performance without relying on ground-truth velocity maps or differentiable forward operators. This approach provides a flexible solution for FWI, offering new possibilities in real-world settings.

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

APPENDIX

# A    USE OF LARGE LANGUAGE MODELS (LLMs)

In preparing this manuscript, we utilized a large language model (LLM) to assist with the writing process. Its role was limited to improving language quality, including grammar, phrasing, and overall readability, as well as helping with LaTeX formatting for tables and equations. The conception of the research, design of experiments, analysis of results, and all scientific contributions were the responsibility of the authors.

# B    COMPARISON WITH PREVIOUS FWI METHODS FROM A GRADIENT PERSPECTIVE

To elucidate the relationship among supervised, self-supervised with a differentiable forward operator, and reinforcement learning based self-supervised approaches, we provide a unified analysis from the perspective of gradient construction and propagation.

**Supervised learning:**    In the supervised paradigm, a neural network $g_\theta$ maps seismic measurements $p$ to a predicted velocity map $\hat{V} = g_\theta(p)$. Training relies on paired ground-truth velocity maps $v$, with a loss function of the form

$$\mathcal{L}_{\text{sup}}(\theta) = \mathbb{E}_{(p,V)\sim\mathcal{D}} \| V - g_\theta(p) \|, \tag{9}$$

where $\mathcal{D}$ denotes the joint distribution of seismic data and corresponding velocity maps, and $\| \cdot \|$ denotes a generic norm (e.g., $\ell_1$, $\ell_2$, or mixed/perceptual norms). The gradient is obtained via direct backpropagation:

$$\nabla_\theta \mathcal{L}_{\text{sup}} = \mathbb{E}_{(p,V)\sim\mathcal{D}} \frac{\partial \mathcal{L}_{\text{sup}}}{\partial \hat{V}} \cdot \frac{\partial \hat{V}}{\partial \theta} = \mathbb{E}_{(p,V)\sim\mathcal{D}} \underbrace{\frac{\partial \mathcal{L}_{\text{sup}}}{\partial g_\theta(p)}}_{\text{Ground truth involved}} \cdot \frac{\partial g_\theta(p)}{\partial \theta}, \tag{10}$$

where the learning signal is explicitly anchored to the availability of ground-truth velocity maps. While this yields stable optimization and strong supervision, it is impractical in real-world scenarios due to the scarcity of paired data.

**Self-supervised learning with a differentiable forward operator:**    The UPFWI framework (Jin et al., 2022) removes the dependence on ground truth by incorporating a differentiable forward operator $f$ that simulates seismic data from predicted velocity maps:

$$\hat{V} = g_\theta(p), \quad \tilde{p} = f(\hat{V}). \tag{11}$$

The reconstruction objective is defined as

$$\mathcal{L}_{\text{diff}}(\theta) = \mathbb{E}_{p\sim\mathcal{P}}\|p - \tilde{p}\| = \mathbb{E}_{p\sim\mathcal{P}}\|p - f(g_\theta(p))\|, \tag{12}$$

with gradients computed via the chain rule:

$$\nabla_\theta \mathcal{L}_{\text{diff}} = \mathbb{E}_{p\sim\mathcal{P}} \frac{\partial \mathcal{L}_{\text{diff}}}{\partial \tilde{p}} \cdot \frac{\partial \tilde{p}}{\partial \hat{V}} \cdot \frac{\partial \hat{V}}{\partial \theta} = \mathbb{E}_{p\sim\mathcal{P}} \frac{\partial \mathcal{L}_{\text{diff}}}{\partial f(g_\theta(p))} \cdot \underbrace{\frac{\partial f(g_\theta(p))}{\partial g_\theta(p)}}_{\text{Differentiable}} \cdot \frac{\partial g_\theta(p)}{\partial \theta}. \tag{13}$$

This formulation enables end-to-end training using only seismic data, yet a differentiable forward operator is required to compute $\partial f/\partial \hat{V}$.

**Self-supervised learning via reinforcement learning:**    Our proposed DeepWaveRL relaxes the differentiability constraint by reframing FWI as a policy optimization problem. With the definition

in the above sections, we can derive the gradient of our objective as follows (clipping is omitted for brevity):

$$\nabla_\theta \mathcal{J}(\theta) = \nabla_\theta \mathbb{E}_{p \sim \mathcal{P}, \{\hat{V}\}_{i=1}^G \sim \pi_{\theta_{\text{old}}}(\cdot|p)} \frac{1}{G} \sum_{i=1}^G \left\{ m_i(\theta) \hat{A}_i \right\} \tag{14}$$

$$= \mathbb{E}_{p \sim \mathcal{P}, \{\hat{V}\}_{i=1}^G \sim \pi_{\theta_{\text{old}}}(\cdot|p)} \left[ \frac{1}{G} \sum_{i=1}^G m_i(\theta) \, \hat{A}_i \cdot \nabla_\theta \log m_i(\theta) \right] \tag{15}$$

$$= \mathbb{E}_{p \sim \mathcal{P}, \{\hat{V}\}_{i=1}^G \sim \pi_{\theta_{\text{old}}}(\cdot|p)} \tag{16}$$

$$\left[ \frac{1}{G} \sum_{i=1}^G \left( \frac{\pi_\theta(\hat{V}_i \mid p)}{\pi_{\theta_{\text{old}}}(\hat{V}_i \mid p)} \right)^{\frac{1}{H \cdot W}} \underbrace{\hat{A}_i}_{\text{Forward model only involved as a multiplier}} \cdot \frac{1}{H \cdot W} \sum_{h,w} \nabla_\theta \log \pi_\theta(\hat{v}_{i,h,w} \mid p) \right]. \tag{17}$$

Unlike supervised and differentiable self-supervised learning, where the error must be differentiable with respect to $\hat{V}$ and thus couples optimization tightly to the properties of $f$, reinforcement learning replaces this requirement by transforming non-differentiable errors into reward signals that directly reweight policy gradients.

Thus, the three paradigms can be interpreted within a common gradient-based framework: **Supervised:** Gradients are directly computed from discrepancies with labeled velocity maps, yielding stable optimization but requiring costly ground truth. **Differentiable self-supervised:** Gradients originate from seismic reconstruction error and backpropagate through a differentiable forward operator, enabling physics-informed learning but restricted by differentiability and high computational cost. **RL-based self-supervised:** Gradients arise from log-likelihood weighting in policy space, with seismic misfit entering only as a reward. This bypasses differentiability, accommodates arbitrary forward operators, and enables stochastic exploration in complex inversion landscapes.

## C  TECHNICAL DETAILS

We normalize the input seismic data to the range $[-1, 1]$ and apply the logarithm transformation with $k = 3$ and $c = 0$ on seismic data. For optimization, we employ the AdamW optimizer with momentum parameters $\beta_1 = 0.9$, $\beta_2 = 0.999$, and a weight decay of $1 \times 10^{-4}$ to update all parameters of the network. The details of hyperparameters and training settings are provided in Table 4. The $\epsilon_{\text{low}}$ and $\epsilon_{\text{high}}$ are 0.2 and 0.27, respectively. For the network architecture, we adopt a four-layer encoder-decoder Vision Transformer (ViT), and we append four convolutional blocks with upsampling layers ($5\times$ and $2\times$), batch normalization, and leaky ReLU as activation functions to map the output of the decoder to $70 \times 70$ velocity map with 100 bins. We implement our models in Pytorch and train them on 16 NVIDIA H100 GPUs.

| | CVA | | FFA & CFA | |
|---|---|---|---|---|
| Test-time Optimization | | ✓ | | ✓ |
| Training Steps | 44,880 | 1,440 | 7,360 | 1,600 |
| Initial Learning Rate | 8e-4 | 1.6e-4 | 6.4e-4 | 6.4e-4 |
| Learning Rate Decay | / | 1,360 | / | / |
| Batch Size | 128 | 2048 | 2048 | 2048 |
| Group Size | 256 | 16 | 32 | 32 |

Table 4: Training details

## D  VISUALIZATIONS

### D.1  MORE VISUALIZATION RESULTS ON CURVEVEL-A

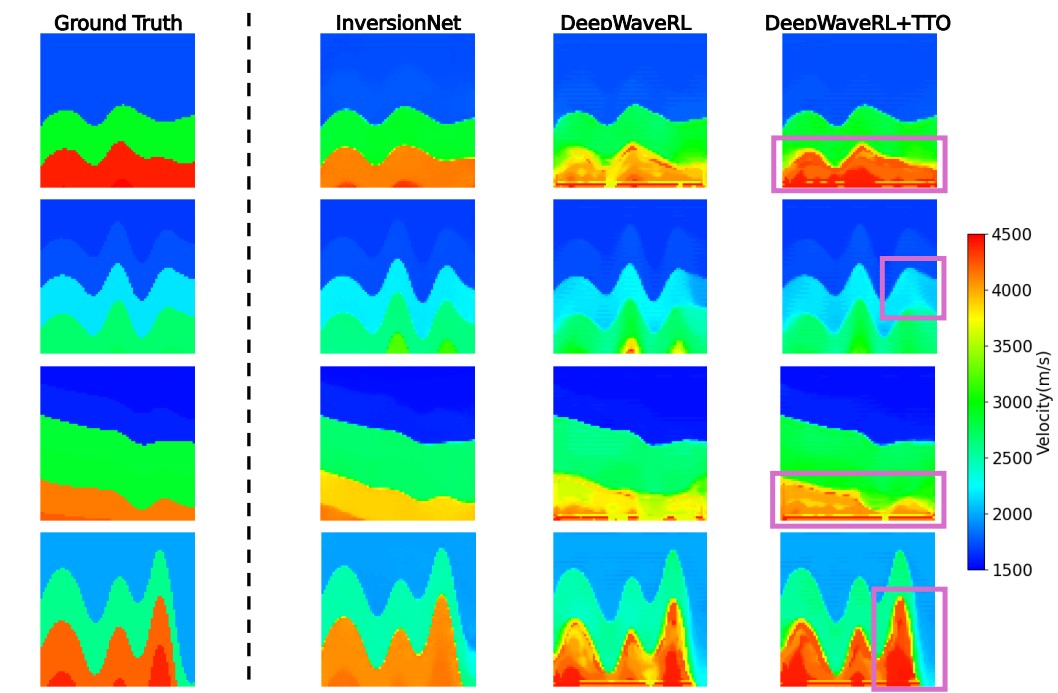

Figure 6: Illustration of ground truth and inversion results of different methods on CurveVel-A.

## D.2 MORE VISUALIZATION RESULTS ON FLATFAULT-A AND CURVEFAULT-A

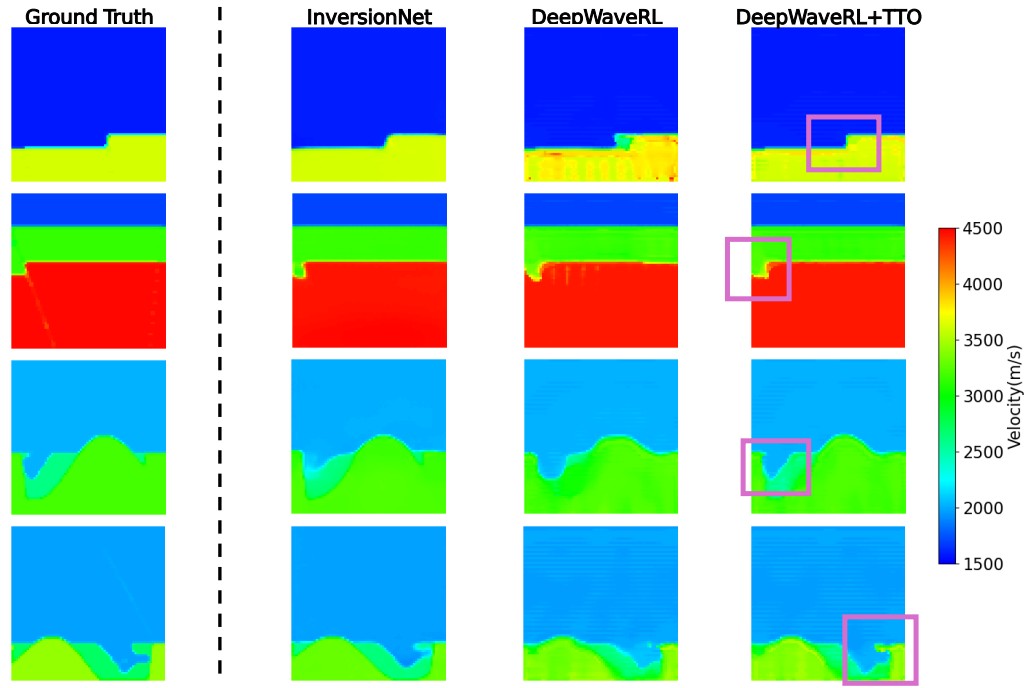

Figure 7: Illustration of ground truth and inversion results of different methods on FlatFault-A and CurveFault-A.

### D.3 VISUALIZATION RESULTS OF DEEPWAVERL WITHOUT LOGARITHM TRANSFORMATION

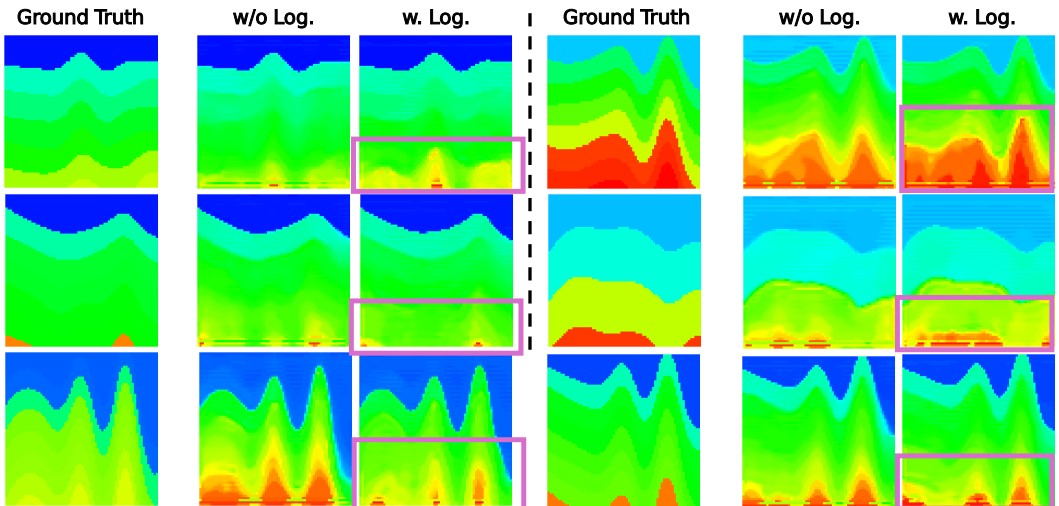

Figure 8: Illustration of ground truth and inversion results of DeepWaveRL with and without the sign-preserving logarithm transformation

### D.4 FAILURE CASES

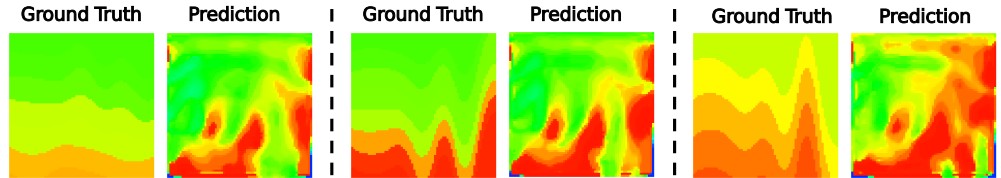

Figure 9: Examples of failure cases where the predictions collapse to similar patterns.

### D.5 EXAMPLES OF UNSTABLE TRAINING RESULTS

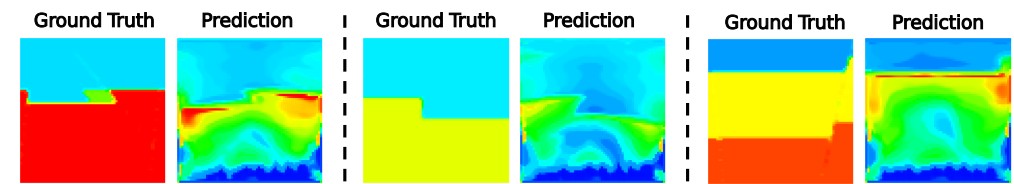

Figure 10: Examples of predicted velocity maps generated by the model that has experienced unstable training.

