# OpenReview forum: "DeepWaveRL: Self-Supervised Full Waveform Inversion via Reinforcement Learning"
_ICLR.cc/2026/Conference — ICLR 2026 Conference Withdrawn Submission_

### Official Review · Reviewer_HxMq · 2025-10-28

**Soundness:** 2
**Presentation:** 3
**Contribution:** 2
**Rating:** 4
**Confidence:** 4

**Summary:**

This paper introduces DeepWaveRL, a reinforcement learning (RL)-based self-supervised framework for Full Waveform Inversion (FWI).
The authors reformulate FWI as a policy optimization problem, where a neural network predicts velocity maps as actions, and the forward modeling operator (possibly non-differentiable) provides scalar rewards based on seismic reconstruction errors. By avoiding gradient backpropagation through the forward operator, DeepWaveRL claims to overcome the “differentiability constraint” that limits prior self-supervised FWI methods such as UPFWI. To stabilize training, the paper proposes three strategies: discretized velocity actions, a sign-preserving logarithmic transformation of seismic data, and policy transfer learning. Experiments on multiple OpenFWI datasets show that DeepWaveRL achieves performance close to supervised methods while requiring no ground-truth velocity labels or differentiable simulators.

**Strengths:**

1. Reformulating FWI as an RL-based policy learning problem is an interesting and unconventional idea, potentially broadening the design space for physics-driven inversion.
2. The method adapts group-based policy optimization (GRPO/DAPO) to spatially structured actions.
3. Results on three OpenFWI datasets (CurveVel-A, FlatFault-A, CurveFault-A) with quantitative metrics and visualizations are thorough.
4. The paper is well organized, clear, and easy to follow, with detailed appendices and visualization of results.

**Weaknesses:**

1. The central motivation claims that self-supervised FWI requires differentiable forward operators, which severely limits applicability.
However, in practice, modern frameworks (e.g., Devito, JAX-FWI, TorchFWI) already support efficient automatic differentiation or adjoint-state gradient computation. Thus, the so-called “differentiability constraint” is not a fundamental bottleneck in current research. The paper should clarify whether the proposed RL-based formulation truly targets industrial black-box solvers or merely offers a conceptual alternative to gradient-based optimization. Without such clarification, the motivation feels weaker than stated.
2. FWI is inherently a continuous regression problem over the velocity field.
Discretizing the velocity range into bins and treating prediction as a classification task introduces quantization artifacts and non-physical discontinuities in the reconstructed velocity maps. The approach implicitly assumes fixed bounds (e.g., 1500–4500 m/s in OpenFWI), which restricts generalization to other datasets or real geological settings where velocity distributions vary. While discretization may stabilize RL training, it compromises physical fidelity and model transferability. The authors should justify this choice more rigorously or explore continuous-action RL alternatives.
3. The “gradient perspective” section restates differences between supervised, self-supervised, and RL-based gradients but does not establish a formal link between the RL objective and the traditional adjoint-state gradient.
There is no theoretical analysis of reward variance, convergence, or sample efficiency.
4. RL-based optimization requires multiple stochastic samples and forward simulations per update, likely increasing computational cost relative to differentiable methods. The paper omits runtime or efficiency analysis.

**Questions:**

1. Reframe the motivation more realistically: focus on scenarios with black-box forward simulators or no adjoint gradients, not on differentiability as a universal limitation.
2. Replace hard discretization with continuous or differentiable quantization policies (e.g., soft binning, Gaussian actions, or continuous actor-critic RL).
3. Provide runtime and efficiency benchmarks against UPFWI and differentiable baselines.
4. Add robustness tests on datasets with different velocity ranges to evaluate generalization.
5. Include theoretical discussion of how policy gradients approximate adjoint-state gradients, or under what conditions the reward function leads to equivalent optimization behavior.

---

### Official Review · Reviewer_r1uT · 2025-10-31

**Soundness:** 2
**Presentation:** 3
**Contribution:** 1
**Rating:** 2
**Confidence:** 4

**Summary:**

The paper applies group-based reinforcement learning to full waveform inversion (FWI), removing the requirement for differentiable forward operators but achieving results substantially worse than supervised baselines. While the motivation to eliminate differentiability constraints is valid, the practical utility is questionable given the massive computational overhead and limited performance gains.

**Strengths:**

Removes differentiability requirement.

**Weaknesses:**

- G=256 forward passes per sample means ~6.1M simulations for 24K training samples plus additional test-time optimization passes. No wall-clock time, GPU-hours, or comparison of compute budgets is provided. For production seismic surveys this would be orders of magnitude too expensive.

- MAE is 29% worse on CurveVel-A (0.0527 vs 0.0409) and 174% worse on FlatFault-A (0.0268 vs 0.0098) compared to supervised methods. Claims of "competitive performance" are misleading when errors nearly double or triple.

- Modern geophysics uses open-source differentiable solvers (Devito, JAX). No evidence supports that black-box/non-differentiable operators are common enough to justify 100x computational overhead. The cited edge cases (fractures, pressure thresholds) are not validated as representative problems.

- Method requires pre-training on another dataset to avoid "unstable learning and convergence issues." This dependency contradicts the self-supervised narrative and limits applicability to novel domains without suitable initialization.

- Continuous action space caused "unstable training and noisy
  predictions," forcing discretization into 100 hardcoded bins. Log
  transform hyperparameters k and c require manual tuning. Does this
  suggest the RL formulation is fundamentally mismatched to the problem?

- Results copied from Deng et al. (2022) without matched compute budgets. Equivalent compute should be allocated to UPFWI (iterative refinement, ensembling) to compare efficiency fairly.

- This is GRPO with sequence -> map substitution (Eq. 4). No
  geophysics-informed reward shaping, physics constraints beyond forward
  operator, or analysis of learned representations. Pure application
  paper that underperforms existing methods. Also, from the application
  point of view, the computational complexity makes it impractical. So,
  I am not sure what the main contribution of this paper is.

- Real-world FWI has strict time/cost budgets. A method requiring 100x
  compute and producing worse results does not enable "new
  possibilities," it's a niche proof-of-concept at best.

**Questions:**

See above

---

### Official Review · Reviewer_EzrU · 2025-10-31

**Soundness:** 3
**Presentation:** 3
**Contribution:** 2
**Rating:** 2
**Confidence:** 4

**Summary:**

The paper explores the application of reinforcement learning (RL) to full waveform inversion (FWI). It uses the fact that in RL, the computation of the advantage function is independent of the policy parameters, allowing the method to avoid differentiating through the forward operator (the solution of the wave equation). The velocity field, i.e., the result of FWI, is the action of the RL agent, while the computed wave profiles are used to compute rewards. The authors also introduce techniques to ensure stable training. The methods performance is demonstrated on simple 2d examples, against other deep learning approaches.

**Strengths:**

* It is an interesting idea to apply RL to FWI, in particular to remove the need to differentiate through the forward operator, compared to other unsupervised learning methods. This allows the usage of faster, non-differentiable forward operators.
* The authors introduce techniques to stabilize training. They conduct ablation studies, demonstrating components that contribute to higher performance.
* The manuscript is easy to read, with a straightforward structure which makes it easy to follow. The figure that compares different FWI methods clearly illustrates the model architectures and data pipelines.
* The authors compare their method to meaningful benchmarks, assuring the credibility of the evaluation.
* Limitations are acknowledged. The authors not only present the impressive results, but also report the cases where the method fails - an important component of the work, adding credibility.

**Weaknesses:**

* The mode collapse seems to be an issue for FWI, the predictions are not just inaccurate, but rather completely wrong.
* The *combination* with FWI is novel, but the SOTA section does not clarify how novel the *methodology* is from an RL perspective. It appears to be a relatively direct application of the GRPO method, with only minor modifications. The authors should improve the SOTA section to clarify this.
* I suspect the training itself is very challenging, given that "adding stability of training" is a major contribution here. The authors did not add a lot of details on challenges in this direction. Adding ablation studies, standard deviation over several training runs with random initial conditions, and duration of training runs would help.
* The paper does not provide a comparison of the training process with other methods, focusing only on the final results. As far as I understand, the main advantage of the proposed method is avoiding differentiation through the forward operator, thus allowing the use of faster yet non-differentiable forward operator, but this advantage may be undermined if the training process is much more challenging than other existing methods.
* There is a similar line of research in nondestructive testing, with publications discussing transfer learning and nondifferentiable solvers. The authors do not use RL, but use a methodology similar to neural ODEs (adjoint method) to use nondifferentiable solvers but still obtain proper gradients for FWI. Both should be cited and discussed.
 - Herrmann, Leon, Tim Bürchner, Felix Dietrich, and Stefan Kollmannsberger. 2023. “On the Use of Neural Networks for Full Waveform Inversion.” Computer Methods in Applied Mechanics and Engineering 415 (October): 116278. https://doi.org/10.1016/j.cma.2023.116278.
 - Chen, Ricky T. Q., Yulia Rubanova, Jesse Bettencourt, and David Duvenaud. 2018. “Neural Ordinary Differential Equations.” NeurIPS Conference 2018, June 19.
* In FWI, it is quite common to work in 3D space. It seems quite challenging for the method to be extended to 3D, as there will be more degrees of freedom in the action space. The authors did not comment on this.
* There is no theoretical analysis of the method, only computational experiments in simple settings. While this is adequate in general for papers in ICLR, it is certainly a weakness. Computational experiments in challenging, realistic settings would outweigh this, for example.

**Questions:**

* How reliable and robust is the method? With stabilization techniques proposed in the paper, how difficult is it to train such a model? Do other methods also encounter severe failure cases?
* How does test-time optimization (TTO) work? It boosts the performance a lot but is not explained.
* What is the computational and memory complexity when the method is used for 3D fields? Would the method still be feasible?

---

### Note · Authors · 2025-12-03

**Comment:**

We sincerely thank the reviewers and the Area Chair for your time, effort and valuable feedback.

**Withdrawal Confirmation:**

I have read and agree with the venue's withdrawal policy on behalf of myself and my co-authors.